# A Multi-Layer Breast Cancer Model to Study the Synergistic Effect of Photochemotherapy

**DOI:** 10.3390/mi14091806

**Published:** 2023-09-21

**Authors:** Magdalena Flont, Elżbieta Jastrzębska

**Affiliations:** 1Faculty of Chemistry, Warsaw University of Technology, Noakowskiego 3, 00-664 Warsaw, Poland; magdalena.flont@pw.edu.pl; 2Center for Advanced Materials and Technologies CEZAMAT, Warsaw University of Technology, Poleczki 19, 02-822 Warsaw, Poland

**Keywords:** breast cancer, cancer-on-a-chip, sequential photochemotherapy, *meso*-tetraphenylporphyrin, three-dimensional (3D) cell culture, multilayered cell model

## Abstract

Breast cancer is one of the most common cancers among women. The development of new and effective therapeutic approaches in the treatment of breast cancer is an important challenge in modern oncology. Two-dimensional (2D) cell cultures are most often used in the study of compounds with potential anti-tumor nature. However, it is necessary to develop advanced three-dimensional (3D) cell models that can, to some extent, reflect the physiological conditions. The use of miniature cancer-on-a-chip microfluidic systems can help to mimic the complex cancer microenvironment. In this report, we developed a 3D breast cancer model in the form of a cell multilayer, composed of stromal cells (HMF) and breast cancer parenchyma (MCF-7). The developed cell model was successfully used to analyze the effectiveness of combined sequential photochemotherapy, based on doxorubicin and *meso*-tetraphenylporphyrin. We proved that the key factor that allows achieving the synergistic effect of combination therapy are the order of drug administration to the cells and the sequence of therapeutic procedures. To the best of our knowledge, studies on the effectiveness of combination photochemotherapy depending on the sequence of the component drugs were performed for the first time under microfluidic conditions on a 3D multilayered model of breast cancer tissue.

## 1. Introduction

According to statistical data from 2018, breast cancers, cancers of the female reproductive system, and female genital organs caused the death of over a million women [1]. Listed diseases are characterized by hormone dependence, phenotypic and molecular heterogeneity, and adaptive mechanisms, making them among diseases with high resistance to treatment [2,3]. Breast cancer is the most frequently diagnosed cancer, and there are two main types of breast cancer: ductal carcinoma (originating in the cells that line the milk duct) and lobular carcinoma (originating in the cells of the mammary glands of the breast lobules) [4].

Breast cancer diagnostics are based on pathomorphological examinations of the collected tissue material and hormonal and molecular analyses [5]. The treatment of breast cancer is complex. Depending on the type of cancer diagnosed, the stage of disease advancement, and the general health of the patient, several therapeutic methods are used in the treatment of breast cancer. These include surgery, radiotherapy, hormone therapy, chemotherapy, molecular therapies, and photodynamic therapy as a relatively new therapeutic approach. Combination therapy, based on the usage of several therapeutic approaches together, deserves special attention because it has been the most effective in the treatment of breast cancer for many years [6].

In most cases, the selection of a therapeutic strategy for breast cancer depends on the tumor’s molecular characteristics. Chemotherapy is always used to treat the so-called “triple-negative” breast cancers, i.e., when the presence of estrogen receptors in the cancer cells is not diagnosed and the expression of the HER2 receptor is not confirmed [7]. Both preoperative and postoperative treatment involve combination chemotherapy, which usually combines the effects of two or three anthracyclines. Less commonly, antimetabolites or alkylating drugs are used [8]. In turn, photodynamic therapy (PDT), i.e., treatment method of cancerous changes based on the use of a photosensitizer (PS), light, and oxygen dissolved in the tissue, is an approach currently rarely used clinically in the treatment of breast cancer. Nevertheless, PDT has great potential in the treatment of breast cancer due to the high selectivity of photosensitizers and the low invasiveness of this therapy [9,10,11]. It is worth investigating the effectiveness of PDT as a supportive therapy in the treatment of breast cancer.

Combination therapy is widely used in many medical fields, and it is based on the use of more than one drug or more than one type of therapy to treat a disease. The use of combination therapy in the treatment of breast cancers can significantly increase the therapeutic effectiveness and reduce the side effects of treatment, e.g., due to the possibility of reducing the doses of drugs [12]. The effect of combination therapy may be additive when the treatment result is the sum of the two individual therapies, or synergistic when the treatment result is greater than the sum of their individual therapies [12]. Nowadays, combining chemotherapy with other treatment methods or the use of combination chemotherapy is the basis of clinical cancer treatment. Chemotherapy can be combined with gene therapy [13], immunotherapy [14], or phototherapy [15]. The synergy achieved thanks to combination therapy allows for a reduction in the dose of the drug and minimizes the side effects of treatment. In addition, it also gives a chance for effective treatment of cancers with a heterogeneous structure such as breast cancer. It was also proven that combination therapy can reverse multi-drug resistance that occurred in cancer cells and was caused by monochemotherapy [16]. However, additional research is needed to optimize the procedures of combination therapies, especially in the case of sequential therapies in which compounds are administered one after the other. It is very important because the sequence of the component drugs administration to the cells may influence the overall therapeutic effect [17].

Three-dimensional (3D) cell culture models that mimic the structure of cancer tissue are essential tools in modern cell engineering. The use of lab-on-a-chip microfluidic systems for cancer research allows cancer cells to be cultured in dynamic conditions, similar to conditions in a living organism [18]. The use of microfluidic systems also gives the opportunity to conduct spatial cell co-cultures in which the interactions between non-malignant and cancer cells, and the interactions of the cells with the extracellular matrix (ECM) are preserved. Many studies indicate that the use of co-culture in in vitro cancer modeling studies is of great importance and advantage over the use of monoculture. Under physiological conditions, cancer tissue consists of both cancer cells and non-malignant cells of the same type: fibroblasts, adipocytes, inflammatory cells including lymphocytes and macrophages, as well as capillaries, and lymphatic vessels. Communication between these cells plays a crucial role in cancer development and progression. In particular, the interaction of cancer cells with fibroblasts capable of differentiating into CAF (cancer-associated fibroblasts) is important [19]. Under in vivo conditions, fibroblasts are an important building element of the cancer microenvironment. Studies have shown that the content of these cells in cancer tissue of patients ranges from 20% to 90% [20]. Many research groups are using co-culture in cancer research. Different ratios of normal and cancer cells in culture are used, between 1:1 and 1:10, to mimic the pathological cancer state [21,22].

A cancer-on-a-chip system can partially mimic tumor physiology in laboratory conditions [23,24]. The use of this type of microsystem may be of crucial importance in research into the effectiveness of combination anti-cancer therapies [25]. In vitro 3D cancer models may also allow the study of sequential therapies, especially how the sequence of delivered therapeutic agents to cancer cells influences the efficiency of the used treatment.

In these studies, we propose a microfluidic model of a three-dimensional cell multilayer that can mimic the structure and microenvironment of breast cancer under in vitro conditions. To develop a 3D cellular model, we used a previously designed microfluidic system with a universal design [26]. In this work, we proved the universality of the developed microfluidic technology, which has the potential to be used in the future in modeling other types of tissues characterized by the layered arrangement of cells (e.g., skin or bladder cancer). We used the developed breast cancer-on-a-chip model to analyze the effectiveness of anti-cancer therapies. We hypothesized that thanks to the use of flow conditions in the microsystem, it would be possible to create a cell multilayer composed of stromal cells and breast cancer cells. We assumed that the developed 3D cell model would imitate the structure of the stroma and parenchyma of cancer, and thus (to some extent) it would reflect the structure of a fragment of cancer tissue. We also assumed that the cell multilayer model would be used to test the effectiveness of chemotherapy, PDT, and combined sequential photochemotherapy in breast cancer treatment. The crucial stage of our work was to examine the influence of the sequence of administration of the component drugs during the combination therapy procedure on its effectiveness.

## 2. Materials and Methods

### 2.1. The Fabrication and Geometry of the Microfluidic System

We designed a hybrid, three-layer (PDMS-PDMS-glass) microsystem for the culture of non-malignant and cancer cells in the form of a three-dimensional (3D) cell multilayer. The microsystem consisted of six culture microchambers, allowing several independent analyses to be performed simultaneously. The microchambers in the microsystem were arranged linearly in two rows and connected by a network of microchannels. The microsystem was made of three layers of transparent materials: the first layer of PDMS with a thickness of about 5 mm, equipped with a longitudinal microchamber and channels with a width of 100 µm; the second layer of PDMS with a thickness of 100 µm, equipped with six holes with a diameter of 2 mm of each; and the third layer of sodium glass with dimensions 76 mm × 25 mm × 1 mm. In order to make a thin layer of PDMS, two PMMA plates with dimensions of 100 mm × 100 mm each were used, one of which had a cavity with dimensions of 60 mm × 40 mm and a depth of 100 µm. The cavity between the plates was filled with a mixture of the non-cross-linked PDMS prepolymer with a cross-linking agent (Dow, Sylgard 184) in a weight ratio of 10:1, using a syringe. PDMS was cross-linked at a high temperature (75 °C, 90 min). The plates were then separated to obtain a 100 µm thick polymer membrane. Holes with a diameter of 2 mm each were made in the membrane with a precision Uni-Core punch, in accordance with the geometry of the pattern. All layers of the microsystem were connected using oxygen plasma, which allowed for the creation of a space for cell culture inside the microsystem. The microsystem was used for the culture of human mammary fibroblast and breast cancer cells. To mimic in vivo cancer conditions, mammary fibroblasts were used as the stroma for cancer cells (Figure 1).

The microfluidic system for the 3D culture of non-malignant and cancer breast cells was produced using micromilling techniques and the PDMS casting technique. A detailed description of the microsystem fabrication was presented in our previous works [26,27]. The distribution of culture microchambers in the designed microsystem is consistent with the distribution of wells on a standard 384-well plate. Thanks to that, it was possible to perform spectrofluorometric measurements in the microsystem using a standard multi-well plate reader.

### 2.2. Cell Lines

Non-malignant and cancer breast cell lines were used in our study. The non-malignant cell line was human mammary fibroblasts (HMF), obtained from ScienCell Research. HMF are connective tissue cells that derive from mesoderm. The HMF cells were cultured using 75 cm^2^ standard culture flasks, with surface modified with poly-l-lysine in a concentration of 2 µg/cm^2^ (ScienCell, Carlsbad, CA, USA) in Fibroblast Medium (ScienCell) supplemented with 10% vol. fetal bovine serum (ScienCell), 1% vol. streptomycin and penicillin (ScienCell), and 1% vol. Fibroblast Growth Supplement (ScienCell). The cancer cell line used in this research was human breast adenocarcinoma (MCF-7), obtained from the American Type Culture Collection. The cancer cells were cultured using 25 cm^2^ standard culture flasks in DMEM medium (Biowest, Nuaillé, France), supplemented with 10% vol. fetal bovine serum (Biowest), 1% vol. streptomycin and penicillin (Biowest), and 2 mM L-glutamine (Biowest). Both cell lines were cultured at 37 °C in a humidified atmosphere, including 5% CO_2_ (HeraCell 150, Thermo Scientific, Waltham, MA, USA).

In the microfluidic system, two types of cells were cultured simultaneously in the form of a co-culture. In all studies in the microsystem, the Fibroblast Medium (ScienCell) supplemented with 10% vol. fetal bovine serum (ScienCell), 1% vol. streptomycin and penicillin (ScienCell), and 1% vol. Fibroblast Growth Supplement (ScienCell) was used. A culture medium with a richer composition was used for cell co-culture.

### 2.3. The Cell Introduction into the Microsystem and 3D Cell Culture Creation

Before starting cell culture, the microsystem was sterilized. For this purpose, an ethyl alcohol solution with a concentration of 70% vol. was introduced into the microsystem using a peristaltic pump for 20 min (flow rate = 5 µL/min). At the same time, the microsystem was exposed to UV radiation (UV Black Ray lamp, time = 20 min, a distance of the microsystem from the light source = 20 cm). The microsystem was filled with the medium and incubated for 3 h (37 °C, 5% CO_2_).

To prepare the cell suspension for examination in the microsystem, fibroblasts and cancer cells were washed with DPBS solution (Biowest) and then detached from the culture bottle with trypsin (Biowest). The cells were centrifuged and then resuspended in 1 mL of medium. Next, the cell suspension density was established based on calculations in Thoma chambers and diluted to obtain the appropriate cell density. Cell suspensions with a density of 10^6^ cells/mL (in the case of non-malignant cells) or 3 × 10^6^ cells/mL (in the case of cancer cells) were introduced into the microsystem. The ratio of the seeding density of fibroblasts and cancer cells in the microsystem was optimized. The results of the optimization tests are presented in the Appendix A.

A fibroblast suspension was prepared and introduced into the microsystem via a peristaltic pump at a flow rate of 10 µL/min. The regular cell distribution in the microchambers was monitored with an inverted microscope. Non-malignant cells in the microsystem were incubated for 24 h (37 °C, 5% CO_2_). Once the fibroblasts adhered to the substrate, a cancer cell suspension was prepared. Breast cancer cells were introduced directly onto the layer made of non-malignant cells (flow rate = 10 µL/min) (Figure 2). Some of the cancer cells adhered to the substrate, and some to the layer of non-malignant cells. In this way, a multilayer cell culture was obtained in the microsystem. In the following days of culture, the proliferation and changes in the morphology of the cells in the microsystem were monitored. The medium was changed every 24 h (max. flow rate = 2 µL/min). The obtained cell multilayers were fluorescently stained and imaged using confocal microscopy to confirm the uniform and reproducible cell distribution in all culture microwells in the microsystem. The imaging results are presented in the Appendix A.

The geometry of the microsystem allowed the cells to be placed in the culture microwell. The non-malignant cells introduced into the microsystem stayed at the bottom of the microwells and adhered to the substrate there. The cancer cells also stopped inside the microwells and adhered to a monolayer made of fibroblasts (Figure 2). The cancer cells in the microsystem could proliferate and grow on top of each other.

It was assumed that the formation of a cell multilayer in the microsystem was possible due to interactions between cancer cells and stromal cells (fibroblasts). The fibroblasts used in the experiments are the primary cell lines that are sensitive to some mechanical factors. Direct exposure of the monolayer of non-malignant cells to the flow of the cancer cell suspension could cause fibroblasts to detach from the substrate (due to their strong affinity for cancer cells) [27]. The cell culture in the microwells could reduce the effect of the detachment of the cells. This was an additional benefit resulting from the 3D geometry of the microsystem. The longitudinal shape of the microchambers (above the microwells) allowed the internal volume of the microsystem to be increased. As a result, additional space above the cell multilayer, filled with the culture medium was obtained. Thus, the geometry of the microsystem was designed to provide the cells with a sufficient culture medium to maintain the normal vital functions of the cells in a multilayer culture.

Studies on the proliferation of normal cells and cancer breast cells growing in the form of cell monolayers were also performed in the developed microsystem. In order to obtain monolayers of MCF-7 and HMF cells, cell suspensions with densities of 3 × 10^6^ cells/mL and 10^6^ cells/mL, respectively, were introduced into the microsystem. Then, the microsystem with cells was incubated for 24 h.

### 2.4. Analysis of 3D Cell Culture Formation under Microfluidic Conditions

In order to confirm the three-dimensional, spatial, and multilayered arrangement of cells grown in the microsystem, the non-malignant and cancer cells were stained with fluorescent dyes (CellTrackers, Thermo Fisher Scientific, Waltham, MA, USA). The non-malignant cells (HMF) were stained with red CMTPX dye. For this purpose, the cells in the culture vessel were incubated with 1 mL of CMTPX solution at a concentration of 6.25 µg/mL for 45 min (37 °C, 5% CO_2_). MCF-7 cells were stained with CMFDA, which showed green fluorescence. For this purpose, the cancer cells were incubated with 1 mL of CMFDA dye solution at a concentration of 5 µg/mL, also for 45 min (37 °C, 5% CO_2_). CellTracker dye working solutions were prepared via 1000-times dilution of the stock of CMFDA and CMTPX solutions in a culture medium without FBS and phenol red. The stained cells were detached from the surface, then introduced into a microsystem and cultured for 96 h. After incubation, a Z-axis scan of cell culture was performed using a confocal microscope (Zeiss Axio Observer 7 with LSM 900). Image acquisition was performed using ZEN Blue 3.6 software.

### 2.5. Analysis of Cell Proliferation in Co-Culture

#### 2.5.1. AlamarBlue Assay

The AlamarBlue assay was used to test the proliferation and determine the metabolic activity of cells grown in the microfluidic system. For this purpose, a 10% vol. AlamarBlue reagent solution was prepared in a culture medium. The solution was loaded into the microsystem using a peristaltic pump (for 15 min, flow rate = 2 µL/min). The microsystem with the cells was incubated for 45 min (37 °C, 5% CO_2_), and then the fluorescence intensity (λex = 552 nm, λem = 582 nm) was measured with a plate reader (Varian, Palo Alto, CA, USA). The time of cell incubation with the AlamarBlue reagent solution was selected experimentally. The measured fluorescence intensity was proportional to the number of metabolically active (viable) cells in the population.

#### 2.5.2. Flow Cytometry

Changes in the number of non-malignant and cancer cells co-cultured in the macroscale for 4 following days were analyzed via flow cytometry. For analysis, the cells were stained with vital fluorescent dyes (CellTrackers, Thermo Fisher Scientific) according to the manufacturer’s instructions. Fibroblasts were stained with CMFDA (green fluorescence). For this purpose, the fibroblasts in the culture vessel were incubated with 1 mL of CMFDA dye solution at a concentration of 5 µg/mL for 45 min (37 °C, 5% CO_2_). The cancer cells were stained with red CMTPX dye. For this purpose, the cells in the culture vessel were incubated with 1 mL of CMTPX solution at a concentration of 6.25 µg/mL, also for 45 min (37 °C, 5% CO_2_). The stained cells were seeded in a 12-well plate. Only non-malignant cells (fibroblasts monoculture) were seeded in the first four wells, only cancer cells (cancer monoculture) were plated in the next four wells, and both non-malignant and cancer cells (co-culture) were co-cultured in the last four wells of the plate. Three different samples were prepared daily for analysis: fibroblast monoculture, cancer cell monoculture, and non-malignant and cancer cell co-culture. The suspensions of the cells detached from the substrate were centrifuged (3 min, 2000 rpm), and each of the obtained cell pellets was resuspended in 300 µL of the medium. Changes in the number of cells in the non-malignant, cancer, and co-culture populations were analyzed using a flow cytometer (CytoFLEX). Sample preparation procedures and analysis were repeated for four following days.

### 2.6. Combined Therapy Procedure in the Microsystem

The combination therapy was performed in sequential mode, which means that the anti-tumor compounds were administered to the cells one after the other. Two sequences of compound administration were used in the research: (1) PDT→DOX, which means that in the first stage the PDT procedure with the use of free *meso*-tetraphenylporphyrin (as photosensitizer—PS) was carried out, and next the chemotherapy procedure with the use of doxorubicin was performed; (2) DOX→PDT, which means that the chemotherapy procedure was performed first, and then the administration of PS solutions and irradiation were performed (PDT procedure).

To perform the combination therapy in the PDT→DOX sequence, photosensitizer solutions with concentrations of 0 µM–10 µM were prepared in the culture medium without FBS and introduced into the microsystems (for 15 min, at a flow rate of 2 µL/min). The cells were incubated with PS for 24 h (37 °C, 5% CO_2_). Next, the medium with the non-accumulated photosensitizer was removed by introducing PBS into the microsystem. Then, a fresh medium was introduced into the microsystem (5 min, 2 µL/min). The cells in the microsystem were irradiated (10 min, λ = 640 nm, 40 mW/cm^2^), and then re-incubated for 24 h (37 °C, 5% CO_2_). In the next stage, doxorubicin solutions with concentrations 0 µM, 1 µM, and 3 µM were prepared in the culture medium without phenol red. Then, the solutions of DOX were introduced into the microsystems (15 min, 2 µL/min). The cells in the microsystem were incubated with doxorubicin solutions for 24 h (37 °C, 5% CO_2_). For the combined therapy in the DOX→PDT sequence, first doxorubicin solutions were introduced into the microsystem (15 min, 2 µL/min), and then incubated for 24 h (37 °C, 5% CO_2_). After this time, PS solutions were introduced (15 min, 2 µL/min). After 24 h incubation (37 °C, 5% CO_2_) the cells were irradiated (10 min, λ = 640 nm, 40 mW/cm^2^). Finally, the cells were re-incubated for 24 h (37 °C, 5% CO_2_). After the combination therapy procedure in the microsystem (PDT→DOX or DOX→PDT), the AlamarBlue test and the CAM/PI differential staining were performed. The control samples were cells untreated with any compound (no TPP, no DOX) and irradiated or non-irradiated.

### 2.7. Differential Staining with Calcein-AM and Propidium Iodide (CAM/PI)

To determine the viability of the cells cultured in the microsystems, a solution containing PI (final concentration: 2 µg/mL) and CAM (final concentration: 4 µM) was introduced into the microsystem with cultured cells. The prepared dye solution was introduced into the microsystem at a flow rate of 2 µL/min. The cells were incubated with fluorescent dyes for 10 min (37 °C, 5% CO_2_). After incubation, the cells were observed with an inverted fluorescence microscope (Olympus IX71). Live cells showed green fluorescence and dead cells showed red fluorescence.

### 2.8. Statistical Analysis

A minimum of three independent repetitions were performed for each tested concentration of compounds (n_min_ = 3). One repetition means performing the tests in one microsystem (in six culture microwells). The obtained results were averaged, and the standard deviation (SD) was calculated. The ANOVA test was performed, which consisted of a one-step analysis of variance. The level of statistical significance α was set at 0.05 (95% confidence interval). Data marked with an asterisk in the graphs indicate statistically significant differences. All results shown in the graphs are expressed as a percentage of cell viability relative to the control.

## 3. Results

### 3.1. Three-Dimensional Cell Culture Imaging in the Microsystem

In order to confirm the spatial (3D) structure of the developed cell model in the microfluidic system, a confocal laser microscope was used. In this experiment, we presented a cross-section of a multilayer cell culture (co-cultures of non-malignant cells and breast cancer cells, Figure 3).

The microscopic images confirm that a spatial culture (3D) was formed in the microsystem after 96 h of culture. The culture was multilayered, and the thickness of the cell layers was about 40 µm (Figure 3). Imaging using confocal microscopy confirmed that the red-stained breast fibroblasts had reorganized and did not form a homogeneous monolayer on the fourth day of culture. Some of the fibroblasts moved between the cancer cells, which brought the construction of the developed cell model closer to the construction of cancer tissue.

### 3.2. Analysis of Breast Cell Proliferation in a Multilayer Model

The proliferation analysis of non-malignant (HMF) and cancer breast (MCF-7) cells was performed in spatial multilayer co-culture. The obtained results were compared with the proliferation of non-malignant cells and cancer cells in the microsystem, where they grew in the form of a cell monolayer. The state of the cell cultures in the microsystem (proliferation) was analyzed every 24 h with the AlamarBlue assay (Figure 4).

The proliferation analysis of HMF cells monolayer showed that during 96 h of culture in the microsystem, the number of cells in the population did not change significantly. A doubling of the number of fibroblasts in the population during the culture was not observed (Figure 4A). Breast cancer cells that were grown under fluidic conditions in the form of cell monolayer divided intensively. A flattening of the cell growth curve was observed, suggesting a slowdown in the rate of cell growth, probably related to reaching the state of high confluence. The total number of MCF-7 cells in the population on the last day of culture (96 h) increased almost three times compared to the first day of culture (24 h, control) (Figure 4B).

Proliferation analysis of the co-culture of HMF and MCF-7 cells showed that after 96 h, there was a significant increase in the total number of cells in the culture. The total number of non-malignant and cancer breast cells in the population increased almost seven times compared to the state on the first day (24 h) and more than three times compared to the state on the second day (48 h, control) (Figure 4C).

The process of cell multilayer formation in the microsystem started after 48 h; therefore, the comparison of the culture status after 48 h with the culture status on the last day was included in the study. The proliferation intensity of cells grown in co-culture was significantly higher than the proliferation intensity of cells grown as monolayers. The changes in cell morphology in all analyzed cell cultures were also assessed. Differences in the shape and size of fibroblasts and breast cancer cells were analyzed. In the case of monolayer cultures, no significant differences in cell morphology were observed during the experiment (96 h). However, after 96 h in cell co-culture, it was observed that MCF-7 cells covered a layer of fibroblasts. The microscopic images from the last day of culture do not show the longitudinal or star-shaped cells that are characteristic of fibroblasts (Figure 4C).

### 3.3. Analysis of Breast Cell Proliferation in a Multilayer Model—Flow Cytometry

Based on the analysis of the growth curves for HMF and MCF-7 cells cultured in the microsystem (Figure 4), it was proved that there were intensive cell divisions in multilayer culture and a significant increase in the total number of cells in the population. The AlamarBlue assay allowed the assessment of changes in the total number of non-malignant and cancer cells in the co-culture. In the next stage of the study, flow cytometry was used to quantify whether cell co-culture influenced only the division of cancer cells or also increased the proliferation of non-malignant cells (determining which of the types of analyzed cells changed the rate of proliferation). Changes in the number of fibroblasts and cancer cells cultured in the macroscale on 12-well plates for 4 following days were assessed (Figure 5). The analysis of the results showed that the percentage of breast cancer cells in the co-culture of HMF and MCF-7 changed during the experiment. A significant change in the percentage ratio of tumor cells to fibroblasts (84%: 15%) was observed after 24 h of co-culture. The reason for this large change was probably the differences in the population doubling time between the two cell types, which were 45 h for fibroblasts and 24 h for breast cancer cells. After 96 h, the proportion of breast cancer cells was about 94% in the co-culture (Figure 5). The obtained observations were consistent with the results of the AlamarBlue assay obtained in the microsystem (Figure 4).

In conclusion, the analysis of fibroblast proliferation in co-culture with breast cancer cells confirmed the assumption that non-malignant cells stimulate the proliferation of cancer cells. The imitation of intercellular interactions and the mimicking of the tumor stroma under in vitro conditions allowed the cell culture in the form of a three-dimensional cell multilayer (3D culture).

### 3.4. Investigation of the Dependence of the Efficacy of Combine Photochemotherapy on the Sequence of the Administered Compounds

The developed cancer-on-a-chip cell multilayer model was used to evaluate the effectiveness of combined photochemotherapy performed on non-malignant and cancer breast cells. It was assumed that it is possible to obtain a synergistic effect of photochemotherapy in breast cancer treatment. For this purpose, sequential photochemotherapy is based on the combination of PDT with chemotherapy and the use of two anti-cancer compounds with different mechanisms of action. It was also assumed that using a 3D cell multilayer would affect the efficacy of combined photochemotherapy. To evaluate the effectiveness of photochemotherapy on a 3D model of breast cancer, a multilayer co-culture of fibroblasts (HMF) and cancer (MCF-7) breast cells was carried out in the microsystem. Breast cell co-culture was sequentially treated with two anti-cancer drugs (DOX and TPP). In the first step, the photodynamic therapy procedure was performed on the 3D cell model, and next, the chemotherapy procedure was carried out (sequence PDT→DOX) (Figure 6). Cell viability was assessed via the AlamarBlue assay and PI/CAM differential staining. Preliminary studies were carried out based on which two concentrations of the cytostatic agent were selected for combined therapy. The results of the preliminary studies are presented in the Appendix A.

The analysis of the obtained results showed that TPP was not cytotoxic to breast cells in the whole range of the tested PS concentrations. As expected, a decrease in cell viability was observed in the co-culture of HMF/MCF-7 after PDT. Breast cell viability decreased to approximately 60% for the two highest tested PS concentrations, 5 µM and 10 µM (Figure 6A). A further decrease in cell viability was observed after incubation of cells with doxorubicin at a concentration of 1 µM. For 1 µM TPP, cell viability at the level 58.6% ± 3.9% was observed. At the highest concentration of photosensitizer (10 µM), the viability of breast cells decreased to 42.5% ± 5.8%. As expected, after incubation of breast cells with 3 µM of DOX, the therapeutic effect was enhanced, and as a result, even lower cell viability was observed in the culture (Figure 6A). Cell viability assay based on PI/CAM differential staining confirmed the results obtained with AlamarBlue assay (Figure 6B). A large number of dead cells in the culture confirms the synergistic effect of photochemotherapy carried out on the multilayer of breast cells in the microsystem.

Combined therapy in a sequential mode makes it possible to introduce the components of the therapy in any order. Thus, in the next stage of the research, it was tested how changing the sequence of photochemotherapy affects its effectiveness. Therefore, in the second part of the experiment, the photochemotherapy procedure on a multilayer breast cancer model was performed in the reverse sequence, i.e., DOX→PDT. It means that the chemotherapy (drug administration) process preceded the PDT procedure. Cell viability was assessed via the AlamarBlue assay and PI/CAM differential staining (Figure 7).

After chemotherapy with 1 µM of DOX, followed by PDT, no decrease in cell viability was observed in the co-culture of MCF-7 and HMF cells. For all concentrations of photosensitizer, cell viability in the multilayer was about 100% both before and after the irradiation step (Figure 7A). The use of a higher concentration of the cytostatic (3 µM) followed by the PDT procedure caused a decrease in the viability of breast cells. The viability of breast cells in the multilayer, independently from the tested TPP concentration, was approximately 70% (Figure 7A). The observed decrease in the viability of breast cells was similar to the effect of monochemotherapy, which was performed with the same DOX concentration in the macroscale (Appendix A). Therefore, we supposed that the decrease in viability did not result from the use of photochemotherapy, but only from the toxic effect of the cytostatic. After staining of the cells in the microsystem with PI/CAM, a large number of alive cells was observed (Figure 7B). Differential staining confirmed the high viability of co-cultured non-malignant and cancer breast cells after the combined photochemotherapy procedure in the DOX→PDT sequence.

## 4. Discussion

In our research, we developed a 3D cancer-on-a-chip model that could be used to analyze the effectiveness of anti-cancer therapies. A multilayer cell model was used to test the efficacy of combined photochemotherapy in the treatment of breast cancer. The proliferation of co-cultures of non-malignant and cancer breast cells under microfluidic conditions was investigated in the first stage of this research. Fibroblasts are physiological components of the stroma of a tumor. In our study, it was proven that the presence of fibroblasts in cell culture was a factor that stimulated the division of cancer cells. As a result, cell co-culture under microfluidic conditions allowed us to obtain a spatial cell model similar to the structure of in vivo cancer tissue. The developed microfluidic system combines three features that determine the creation of a 3D multilayer cell culture:Specific geometry that allows for cell culture in a microwell with indirect exposure to flow;Possibility of generating laminar flow (imitating blood flow) and controlling it;Possibility of seeding layers of stroma cells (fibroblasts) and cells imitating tumor parenchyma (cancer cells) in microchambers.

The developed microfluidic system allowed easy observation of reorganization and 3D cell structure (among others thanks to the use of transparent construction materials—PDMS/glass).

Photochemotherapy is a form of combined anti-cancer treatment and it includes the use of light and photosensitizers in therapy, as well as other anti-cancer drugs, e.g., cytostatics [28,29]. In our research, we used two types of compounds with anti-cancer potential and different mechanisms of action. Doxorubicin is a phase-specific anthracycline clinically used in monochemotherapy or combination chemotherapy for many types of cancer, including gynecological and breast cancers [30]. Meso-tetraphenylporphyrin is a synthetic heterocyclic porphyrin compound and a photosensitizer clinically used in photodynamic anti-cancer therapy (PDT) [31]. The PDT mechanism is based on the activation of the photosensitizer accumulated in cells by the light of appropriate power and wavelength. The activated photosensitizer undergoes a series of energy transformations, and as a result, reactive oxygen species are produced in the cancer cell. The effect of PDT is the death of cancer cells [32].

The analysis of the results obtained in our study proved that using two drugs with varying mechanisms of action in the sequential combined therapy of breast cancer could be very effective. However, it was confirmed that the effect of sequential photochemotherapy with doxorubicin and meso-tetraphenylporphyrin significantly depends on the sequence of therapeutic procedures. A synergistic therapeutic effect was observed in the case of using the PDT→DOX sequence. The cumulative effect of the combination of the two therapies was higher than the efficacy of monochemotherapy or photodynamic therapy. It is probable that the accumulation of photosensitizer in breast cancer cells, followed by the action of intracellular oxidative stress induced by TPP activation, sensitized breast cells to the cytostatic effects. We supposed that breast cancer cells became susceptible to the action of doxorubicin, and the replication processes of cancer cells were inhibited in cell culture. However, no synergistic effect was observed when the reverse sequence of component procedures was used during photochemotherapy (DOX→PDT). In addition, the high viability of breast cells after photochemotherapy indicated that only the cytostatic compound influenced their viability. The uptake of the photosensitizer via doxorubicin-loaded breast cancer cells was probably reduced. The accumulation of cytostatics in the cancer cells could reduce the accumulation of the photosensitizer [33]. Thus, the photodynamic therapy procedure did not enhance the therapeutic effect because the photocytotoxic reactions could not proceed correctly, with sufficient efficacy to reduce cell viability.

There are several reports in the literature that confirm that in vitro studies on the use of the combination of PDT with chemotherapy in the treatment of breast cancers, cancers of the female reproductive system, and female genital organs are conducted [34,35,36,37,38]. Benito-Miguel et al. [34] proposed the use of cytostatic nanoparticles to enhance the photodynamic effect during PDT on cervical cancer cells (HeLa). The therapy was performed with 5-fluorouracil encapsulated in biodegradable chitosan nanocapsules in combination with a photosensitizer and 5-aminolevulinic acid. Cancer cell viability, apoptotic processes, and ROS level were examined after sequential combination therapy. The authors proved a synergistic effect of the tested combination therapy. Sanchez-Ramirez et al. [37] studied the efficacy of photochemotherapy on SCOV-3 ovarian cancer cells. Poly (lactide-glycol) nanoparticles combined with carboplatin (cytostatic) and indocyanine green (photosensitizer) were designed and synthesized. An increase in the photocytotoxic effect was observed after irradiation of ovarian tumor cells, which contained accumulated component drugs.

The dependence of the effectiveness of combination therapy on the sequence of the administered compounds was examined [29,39,40,41,42,43,44]. Rizvi et al. [29] studied the sequential combination of the benzoporphyrin monoacid as a photosensitizer with carboplatin action. They performed research using three-dimensional models of ovarian cancer formed in a hydrogel. Carboplatin was administered to the ovarian cells before or after PDT. A synergistic effect was achieved only for one of the used sequences. Erdem et al. [39] conducted a similar study on SCOV-3 ovarian cells. They tested the effectiveness of the combination of verteporphyrin-based PDT with the action of cationic peptides. The obtained results confirmed that only the delivery of cationic peptides to the cells before the photosensitizer and irradiation allows the synergistic effect of the therapy to be achieved. In turn, Ali et al. [40] performed photochemotherapy based on the combination of doxorubicin or methotrexate with PDT based on aluminum phthalocyanines. Various sequences of therapeutic procedures were tested on cancer cells from the cervix, breast, and brain. It was confirmed that the order of drug administration did not affect the effectiveness of the combination therapy.

In contrast to the literature [29,39,40,41,42,43,44], we present for the first time, a study of the effectiveness of combined photochemotherapy on a three-dimensional, spatial, and heterogeneous cell model that mimics the structure of cancer tissue. To the best of our knowledge, a dependence of the efficacy of combined photochemotherapy on the sequence of the component drugs was observed under microfluidic conditions for the first time. In conclusion, the advanced cancer-on-a-chip cell multilayer model was successfully used to perform combined photochemotherapy on non-malignant and cancer breast cells. A synergistic therapeutic effect was obtained, but only in one of the two tested delivery sequences of the component compounds. It was investigated that the chemotherapy procedure can immunize cancer cells against the effects of photodynamic therapy, possibly as a result of reduced photosensitizer accumulation. Therefore, it was proven that the effect of sequential photochemotherapy is strongly dependent on the sequence of component procedures. Photochemotherapy is a method not widely used to treat breast cancers, cancers of the female reproductive system, and female genital organs. Nevertheless, our research shows that there is a potential for clinical use of photochemotherapy in the treatment of breast cancers and obtaining high treatment effectiveness.

## 5. Conclusions

The aim of this study was to develop a microfluidic and three-dimensional cell culture model, that can mimic the heterogeneous structure of breast cancer under in vitro conditions. The developed cancer-on-a-chip model was successfully used to analyze the effectiveness of anti-cancer therapies against breast cancer cells. We showed that a cell multilayer model composed of stromal and breast cancer cells can be formed under microfluidic conditions. It was confirmed that the developed 3D cell model imitated the stroma and the parenchyma of breast cancer and thus could mimic the structure of a fragment of heterogeneous cancer tissue. The cell multilayer model was successfully used to test the efficacy of combined sequential photochemotherapy on breast cancer. Our work has shown that the sequence of administration of the component drugs during the combined photochemotherapy procedure under microfluidic conditions has a crucial impact on its effectiveness.

## Figures and Tables

**Figure 1 micromachines-14-01806-f001:**
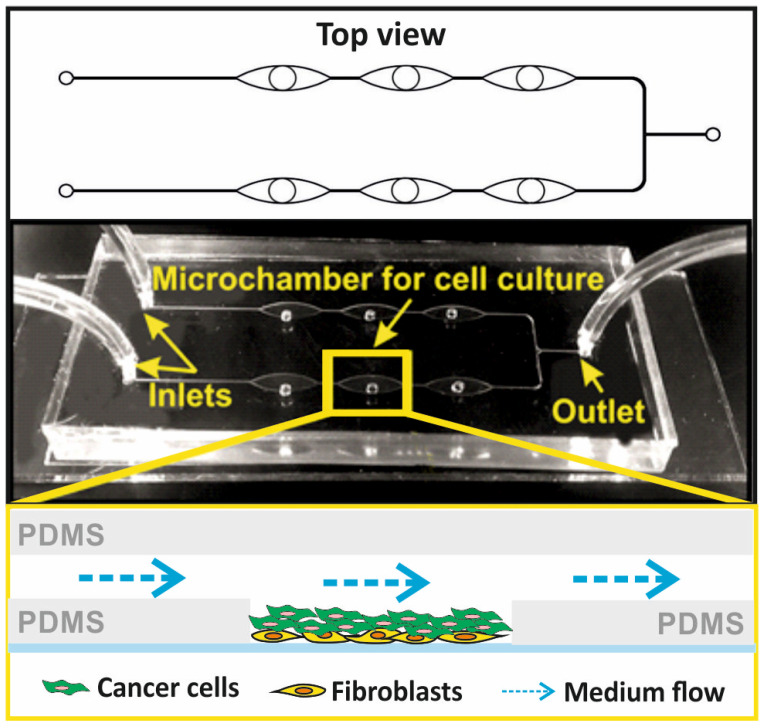
The geometry of the PDMS/PDMS/glass microfluidic system for a spatial multilayer cell culture and a scheme of the cell’s arrangement in the culture chamber.

**Figure 2 micromachines-14-01806-f002:**
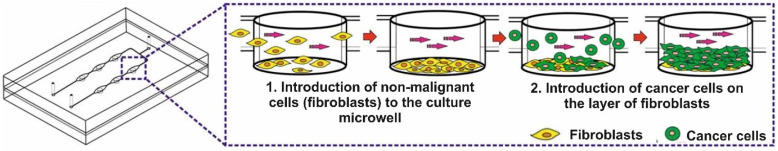
The scheme of cell introduction into the microsystem.

**Figure 3 micromachines-14-01806-f003:**
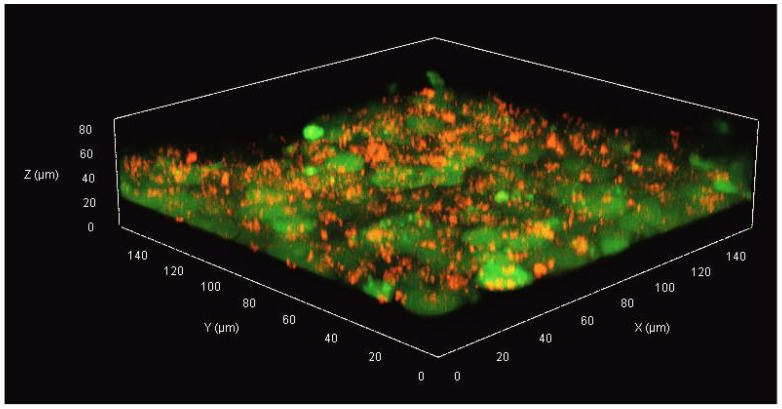
The imaging of three-dimensional cell multilayer in the microsystem. The breast fibroblasts (HMF) were stained with the red fluorescent dye (CMTPX), while the cancer cells (MCF-7) were stained with the green fluorescent compound (CMFDA). The image was acquired on the fourth day of culture (96 h).

**Figure 4 micromachines-14-01806-f004:**
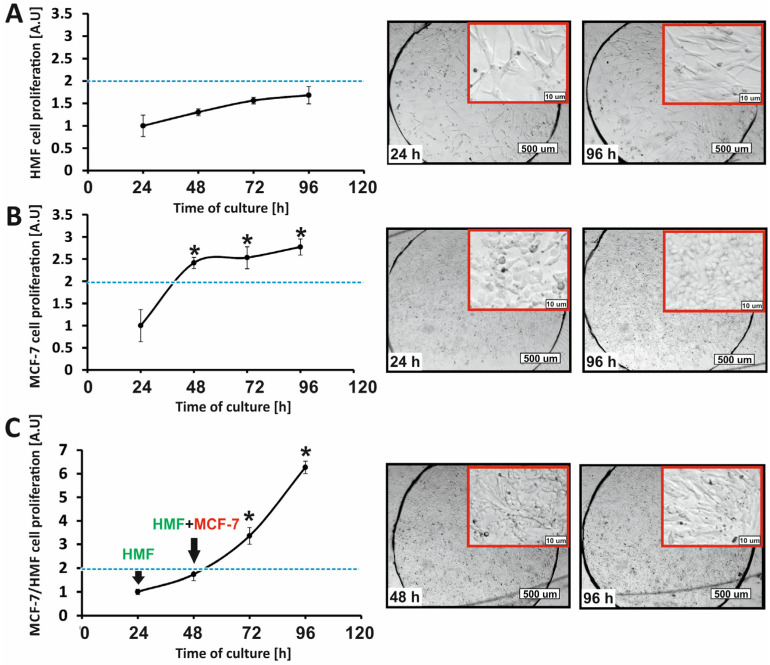
Analysis of the proliferation of non-malignant (HMF) and cancer (MCF-7) breast cells in the microsystem. The intensity of proliferation of (**A**) HMF and (**B**) MCF-7 cell monocultures. (**C**) The intensity of proliferation of co-cultures of HMF and MCF-7 cells (cell multilayer). The images show changes in cell morphology on the first (24 h or 48 h) and last (96 h) day of culture. The blue line marks the doubling of the number of cells in the population (2 times stronger signal). Asterisks indicate data that differ statistically from the control (24 h). The red frames in the microscopic images are close-ups of the cell morphology in the central area of the chamber.

**Figure 5 micromachines-14-01806-f005:**
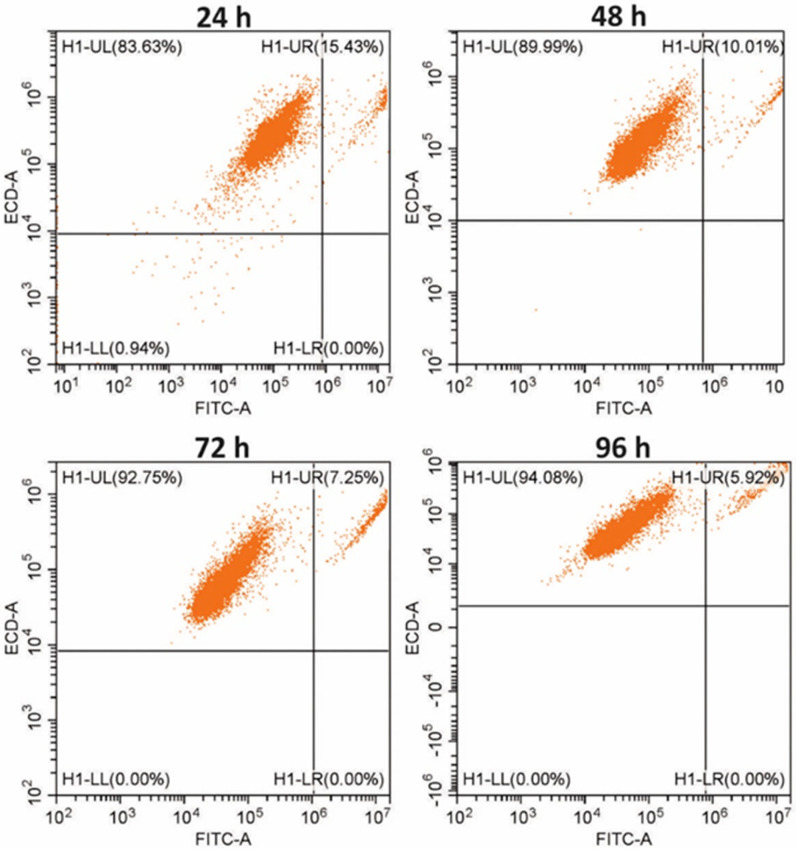
The ratio of the population of cancer cells (MCF-7) and fibroblasts (HMF) in co-culture on the following days of culture. The results were obtained via flow cytometry. First-day ratio of MCF-7/HMF was 50%:50%.

**Figure 6 micromachines-14-01806-f006:**
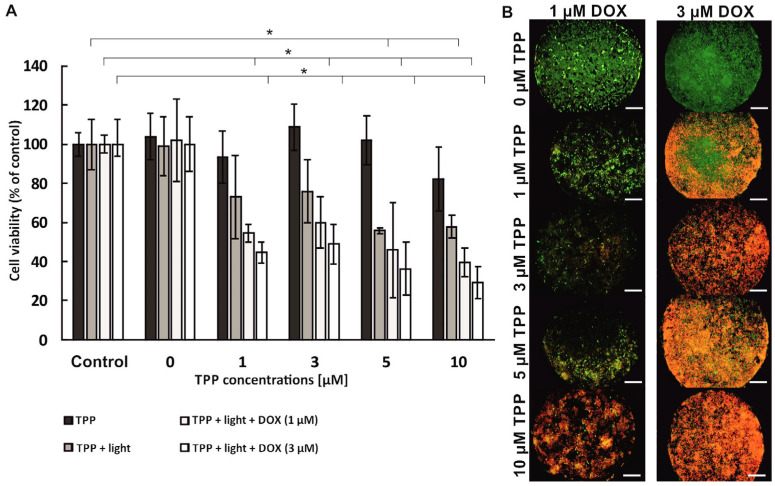
(**A**) Viability of non-malignant (HMF) and cancer (MCF-7) breast cells before and after the photochemotherapy procedure (PDT→DOX). Asterisks indicate statistically significant differences (ANOVA, α = 0.05). (**B**) Microscopic images of HMF/MCF-7 co-culture performed in the microsystem after the photochemotherapy procedure (PDT→DOX) (green cells—alive; red cells—dead). Scale bar: 500 µm.

**Figure 7 micromachines-14-01806-f007:**
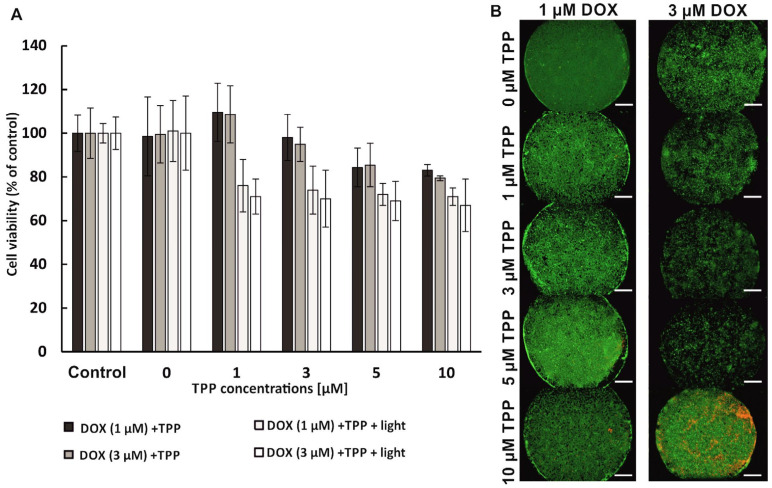
(**A**). Viability of non-malignant (HMF) and cancer (MCF-7) breast cells before and after the photochemotherapy procedure (DOX→PDT). (**B**) Microscopic images of HMF/MCF-7 co-culture performed in the microsystem after the photochemotherapy procedure (DOX→PDT) (green cells—alive; red cells—dead). Scale bar: 500 μm.

## Data Availability

The data presented in this study are available on request from the corresponding author.

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
