# Peer review of "A Multi-Layer Breast Cancer Model to Study the Synergistic Effect of Photochemotherapy"

_micromachines, 2023, doi:10.3390/mi14091806_

Round 1

Reviewer 1 Report

In this manuscript, the authors report on the development of a microfluidic 3D cell culture model that mimics the heterogeneous structure of breast cancer. The developed model was successfully applied to analyze the combined effects of anticancer drug and photodynamic therapy.

However, the principles of this study, including the device geometry, are the same as in the authors' previous paper ref.[27], and no novelty can be found in terms of microdevice research.

The only differences between the two studies are biological: the type of cancer and the combination of photodynamic therapy and anticancer drugs.

Therefore, this manuscript should be submitted to a more specific journal on cancer treatment.

Other points

As shown in the flow cytometry experiments, 3D co-culture can be achieved in a well plate. The advantages of using microfluidic devices should be clearly demonstrated, if any.

Fig. 5 

Flow cytometry data on day zero (50%:50%) should be shown. Also, considering the growth rate of the cells, the change to 84%:15% after 24 h seems too drastic. How do the authors interpret this drastic ratio change?

Fig. 3 

Fluorescence microscopy images after 96 h of co-culture show considerably more fibroblasts, which may not be consistent with the results in Fig. 5.

Fig. 6

Is it possible to show the effects of the combination of photodynamic therapy and anticancer drugs on fibroblasts?

Reviewer 2 Report

This was an interesting paper highlighting the effectiveness of drug treatment on a cancer model in microfluidics using a co-culture.

There are only a few minor problems.

line 30 and 62. Why do the authors mention gynecological cancer here when the manuscript is about breast cancer? I would suggest to change the paragraph.

line 173. Wouldn’t it be better to use an autoclave to sterilize the device?

line 455 and 483. What is the timeline for the drug treatment? How many hours/days did you wait to do the staining?

line 504. “Proved” should be “proven “ 

English was fine

Round 2

Reviewer 1 Report

The manuscript is not perfect, but it has been revised sufficiently.